# LilNetX: Lightweight Networks with EXtreme Model Compression and Structured Sparsification

**Sharath Girish**[1], **Kamal Gupta**[1], **Saurabh Singh**[2] & **Abhinav Shrivastava**[1]

[1]University of Maryland, College Park
{sgirish,kampta,abhinav}@cs.umd.edu

[2]Google Research
saurabhsingh@google.com

## Abstract

We introduce LilNetX, an end-to-end trainable technique for neural networks that enables learning models with specified accuracy-compression-computation trade-off. Prior works approach these problems one at a time and often require post-processing or multistage training. Our method, on the other hand, constructs a joint training objective that penalizes the self-information of network parameters in a latent representation space to encourage small model size, while also introducing priors to increase structured sparsity in the parameter space to reduce computation. When compared with existing state-of-the-art model compression methods, we achieve up to 50% smaller model size and 98% model sparsity on ResNet-20 on the CIFAR-10 dataset as well as 31% smaller model size and 81% structured sparsity on ResNet-50 trained on ImageNet while retaining the same accuracy as these methods. The resulting sparsity can improve the inference time by a factor of almost $1.86\times$ in comparison to a dense ResNet-50 model. Code is available at https://github.com/Sharath-girish/LilNetX.

## 1 Introduction

Recent research in deep neural networks (DNNs) has shown that large performance gains can be achieved on a variety of real world tasks simply by employing larger parameter-heavy and computationally intensive architectures (He et al., 2016; Dosovitskiy et al., 2020). However, as DNNs proliferate in the industry, they often need to be trained repeatedly, transmitted over the network to different devices, and need to perform under hardware constraints with minimal loss in accuracy, all at the same time. Hence, finding ways to reduce the storage size of the models on the devices while simultaneously improving their run-time is of utmost importance. This paper proposes a general-purpose neural network training framework to jointly optimize the model parameters for accuracy, the model size on the disk, and computation, on any given task.

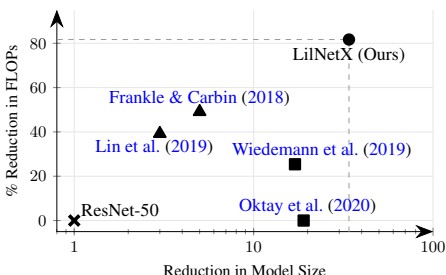

Figure 1: Our method jointly optimizes for size on disk and structured sparsity. We compare various approaches using ResNet-50 architecture on ImageNet and plot FLOPs (y-axis) *vs.* size (x-axis) for models with similar accuracy. Prior model compression methods optimize for either quantization (■) or pruning (▲) objectives. Our approach, LilNetX, enables training while optimizing for both compression (model size) as well as computation (structured sparsity). Refer Table 1 for details.

Over the last few years, research on training smaller and efficient DNNs has followed two seemingly parallel tracks with different goals. One line of work focuses on model compression to deal with storage and communication network bottlenecks when deploying big models or a large number of small models. While they achieve high levels of compression in terms of memory, their focus is not on reducing computation. These works either require additional algorithms with some form of post hoc training (Yeom et al., 2021) or quantize the network parameters at the cost of network performance (Courbariaux et al., 2015; Li et al., 2016). The other line of work focuses on reducing computation through various model pruning techniques (Han et al., 2015; Frankle & Carbin, 2018; Evci et al., 2020). Their focus is to decrease the

number of Floating Point Operations (FLOPs) of the network at inference time, while still achieving some compression due to fewer parameters. Typically, the cost of storing these pruned networks on disk is much higher than dedicated model compression works.

In this work, we bridge the gap between the two lines of work and show that it is indeed possible to train a neural network while jointly optimizing for both the compression to reduce disk space as well as structured sparsity to reduce computation (Fig. 1). We maintain quantized latent representations for the model weights and penalize the entropy of these latents. This idea of reparameterized quantization (Oktay et al., 2020) is extremely effective in reducing the effective model size on the disk. However, it requires the full dense model during the inference. To address this shortcoming, we introduce priors to encourage structured and unstructured sparsity in the representations along with key design changes. Our priors reside in the latent representation space while encouraging sparsity in the model space. More specifically, we use the notion of slice sparsity, a form of structured sparsity where a $K \times K$ slice is fully zero for a convolutional kernel of size $K$ and $C$ channels. Unlike unstructured sparsity which has irregular memory access and offers a little practical speedup, slice-structured sparsity allows for removing entire kernel slices per filter, thus reducing channel size for the convolution of each filter. Additionally, it is more fine-grained than fully structured channel/filter sparsity works (He et al., 2017; Mao et al., 2017) which typically lead to accuracy drops.

Extensive experimentation on three standard datasets shows that our framework achieves high levels of structured sparsity in the trained models. Additionally, the introduced priors show gains even in model compression compared to previous state-of-the-art. By varying the weight of the priors, we establish a trade-off between model size, sparsity, and accuracy. Along with model compression, we achieve inference speedups by exploiting the sparsity in the trained models. We dub our method LilNetX - Lightweight Networks with EXtreme Compression and Structured Sparsification. Our contributions are summarized below.

- We introduce LilNetX, an algorithm to jointly perform model compression and structured sparsification for direct computational gains in network inference. Our algorithm can be trained end-to-end using a single joint optimization objective without any post-hoc training or post-processing.
- With extensive ablation studies and results, we show the effectiveness of our approach while outperforming existing approaches in both model compression and pruning, in most network and dataset setups, obtaining inference speedups in comparison to the dense baselines.

## 2 RELATED WORK

Typical model compression methods usually follow some form of quantization, parameter pruning, or both. Both lines of work focus on reducing the size of the model on the disk, and/or increasing the speed of the network during the inference time, while maintaining an acceptable level of classification accuracy. In this section, we discuss prominent quantization and pruning techniques.

**Model pruning:** A plethora of works show that a large number of network weights can be pruned without significant loss in performance (LeCun et al., 1990; Reed, 1993; Han et al., 2015). Methods such as the Lottery Ticket Hypothesis (Frankle & Carbin, 2018), adapted by various works (Savarese et al., 2020; Frankle et al., 2019; Malach et al., 2020; Girish et al., 2021; Chen et al., 2021; 2020; Desai et al., 2019; Yu et al., 2020) prune models, while reaching the dense network performance, but are iterative and perform unstructured pruning. Other works prune at initialization (Lee et al., 2018; Wang et al., 2020; Liu & Zenke, 2020; Tanaka et al., 2020) and avoid multiple iterations, but show accuracy drops compared to the dense models (Frankle et al., 2020). On the other hand, structured sparsity via filter/channel pruning offers practical speedups at the cost of accuracy (Wen et al., 2016; He et al., 2017; Huang & Wang, 2018). Yuan et al. (2020) obtain almost no drops of network performance with structured sparsification but have lower levels of model compression rates due to storage of floating point weights. Other works operate on intermediate levels of structure such as N:M structured sparsity (Zhou et al., 2021) and block sparsity (Narang et al., 2017). Niu et al. (2020) is the closest to ours in terms of pruning structure utilizing slice sparsity, along with an even finer pattern pruning. They show that such structure can be exploited for inference speedups. They, however, require predefining a filter pattern set and heuristics for determining layerwise sparsity. They also optimize for auxiliary variables and have additional training costs due to the dual optimization subproblem (Ren et al., 2019). In contrast, our algorithm uses a single objective to jointly optimize for sparsity and model compression with very little impact on training complexity.

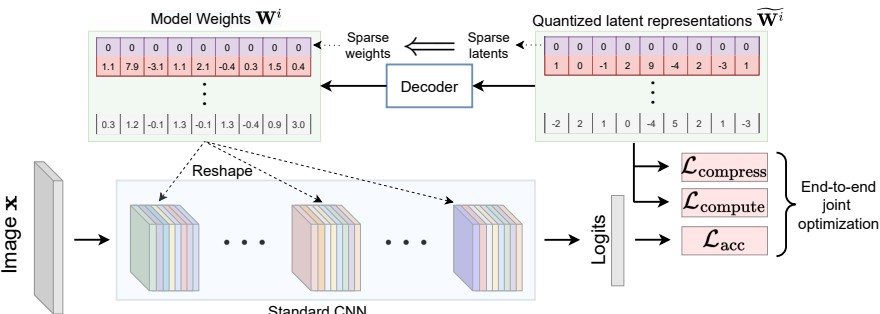

Figure 2: **Overview of our approach.** A standard CNN comprises of a sequence of convolutional and fully-connected layers. We reparameterize the parameters $\boldsymbol{W}^i$ of each of these layers as $\widetilde{\boldsymbol{W}}^i$ in a quantized latent space. Our decoder is such that sparsity in the quantized latents translate to sparsity in CNN parameters. Further, we organize each parameter tensor as a set of slices (depicted as colored bands) corresponding to different channels. Proposed training loss exploits this structure to encourage slice sparsity and jointly optimize for accuracy-compression-computation.

**Model quantization:** Quantization methods discretize the parameters of a network to a small, finite set of values, to store them efficiently using entropy coding methods (Rissanen & Langdon, 1981). Earlier methods uniformly quantize weights to binary or tertiary representations (Courbariaux et al., 2015; Li et al., 2016; Zhou et al., 2018; Zhu et al., 2016; Rastegari et al., 2016; Hubara et al., 2017). Several other works focus on non-uniform Scalar Quantization (SQ) techniques (Tung & Mori, 2018; Zhou et al., 2017; Nagel et al., 2019; Banner et al., 2018; Wu et al., 2016; Zhang et al., 2018; Oktay et al., 2020). Vector Quantization(VQ) (Gong et al., 2014; Stock et al., 2019; Wang et al., 2016; Chen et al., 2015; 2016) on the other hand, is a more general technique, where the representers can take any value. VQ can be done by clustering of CNN layers at various compression-accuracy trade-offs (Faraone et al., 2018; Son et al., 2018), hashing (Chen et al., 2015; 2016), or residual quantization (Gong et al., 2014). Quantization works focus on reducing bit widths (Zhao et al., 2019; Jain et al., 2020) which has the effect of high model compression (Young et al., 2021). A few works do provide inference speedups (Hubara et al., 2016; Jacob et al., 2018) by utilizing lower bit arithmetic but require custom hardware. In comparison, we focus on jointly optimizing for compression and computational gains, and leave optimization for lower precision arithmetic for future. We show the benefits of our approach in terms of even smaller model size compared to these quantization works along with computational gains while maintaining high levels of accuracy in §5.

## 3 APPROACH

We consider the task of classification using a convolutional neural network (CNN), although our approach can be trivially extended to other tasks such as object detection or generative modeling. Given a dataset of $N$ images and their corresponding labels $\{\mathbf{x}_i, y_i\}_{i=1}^N$, our goal is to train a CNN with parameters $\boldsymbol{\Theta}$ that are jointly optimized to: 1) maximize classification accuracy, 2) compress the model by minimizing the number of bits required to store the model on disk, and 3) minimize the computational cost of inference in the model by maximizing model sparsity. To keep our method end-to-end trainable, we formulate it as minimization of a joint objective that allows for an accuracy-compression-computation trade-off as

$$\mathcal{L}(\boldsymbol{\Theta}) = \mathcal{L}_{\text{acc}}(\boldsymbol{\Theta}) + \mathcal{L}_{\text{compress}}(\boldsymbol{\Theta}) + \mathcal{L}_{\text{compute}}(\boldsymbol{\Theta}). \tag{1}$$

For the task of classification, the accuracy term $\mathcal{L}_{\text{acc}}(\boldsymbol{\Theta})$ is the usual cross-entropy loss. Given an image, it maximizes the probability assigned to the target label. The compression term $\mathcal{L}_{\text{compress}}(\boldsymbol{\Theta})$ encourages the model to have a small disk size. We reparameterize the model weights using a quantized latent representation space which are then stored on disk. The compression term, while encouraging smaller model size, doesn't lead to computational gains as the decoded model parameters are still dense. Our computation term $\mathcal{L}_{\text{compute}}(\boldsymbol{\Theta})$ addresses this issue by introducing a structured sparsity-inducing loss. Our framework allows the structured sparsity in latent weight space to directly translate to the structured sparsity in the decoded model weights. In our experiments, we demonstrate significant speedups in the inference time of our model with sparse weights using off-the-shelf libraries. Refer to Fig. 2 for a high-level overview of our approach. In the following sections, we describe the compression and computation terms in more detail.

## 3.1 COMPRESSION TERM

We formulate our compression term by building upon prior works that incorporate entropy penalty on parameter representation during training (Ballé et al., 2018; Oktay et al., 2020). The model's weights and biases, $\boldsymbol{\Theta}$, are reparameterized as quantized latent representations which are compressed while the network parameters are implicitly defined as a transform of the latent representations.

We represent the set of model parameters $\boldsymbol{\Theta} = \left\{ \boldsymbol{W}^1, \boldsymbol{b}^1, \boldsymbol{W}^2, \boldsymbol{b}^2, \ldots, \boldsymbol{W}^N, \boldsymbol{b}^N \right\}$ where $N$ is the total number of layers in the network, and $\boldsymbol{W}^k, \boldsymbol{b}^k$ represent the weight and bias parameters of the $k^{\text{th}}$ layer. Each of these parameters can take continuous values during inference. However, these parameters are stored using quantized latent representations belonging to a corresponding set $\boldsymbol{\Phi} = \left\{ \widetilde{\boldsymbol{W}}^1, \widetilde{\boldsymbol{b}}^1, \widetilde{\boldsymbol{W}}^2, \widetilde{\boldsymbol{b}}^2, \ldots, \widetilde{\boldsymbol{W}}^N, \widetilde{\boldsymbol{b}}^N \right\}$. For each convolutional layer, $\boldsymbol{W}$ is a weight tensor of dimensions $C_{\text{in}} \times C_{\text{out}} \times K \times K$, where $C_{\text{in}}$ is the number of input channels, and $C_{\text{out}}$ is the number of output channels, and $K$ denotes the filter width and height. The corresponding quantized latent representation $\widetilde{\boldsymbol{W}}$ is represented by a two-dimensional (2D) matrix of size $C_{\text{in}} C_{\text{out}} \times K^2$. For each dense layer, $\boldsymbol{W}$ is a tensor of dimension $C_{\text{in}} \times C_{\text{out}}$ and its corresponding latent representation $\widetilde{\boldsymbol{W}}$ is a matrix of dimension $C_{\text{in}} C_{\text{out}} \times 1$. All the biases can be represented in the same way as dense layers. Each latent representation is a quantized 2D matrix $\widetilde{\boldsymbol{W}} \in \mathbb{Z}^{C_{\text{in}} C_{\text{out}} \times l}$ where $l = 1$ for dense weights (and biases) while $l = K^2$ for convolutional weights. Each row/slice $\widetilde{\boldsymbol{W}}_i$ from $\widetilde{\boldsymbol{W}}$ represents a sample drawn from an $l$-dimensional discrete probability distribution. In order to decode parameters from latent space $\boldsymbol{\Phi}$ to model space $\boldsymbol{\Theta}$, learnable affine transforms $\boldsymbol{\Psi}$ are introduced.

$$\boldsymbol{W}_i = \boldsymbol{\Psi}_{\text{scale}} \left( \widetilde{\boldsymbol{W}}_i + \boldsymbol{\Psi}_{\text{shift}} \right), \quad \boldsymbol{W} = \text{reshape} \left( [\boldsymbol{W}_1 \quad \dots \quad \boldsymbol{W}_i \quad \dots] \right), \quad i \in [1 \dots C_{\text{in}} C_{\text{out}}] \quad (2)$$

Where $\widetilde{\boldsymbol{W}}_i$ represents the $i^{\text{th}}$ row/slice of $\widetilde{\boldsymbol{W}}$ and $\boldsymbol{W}_i$ is the corresponding decoded slice. $\boldsymbol{\Psi}_{\text{scale}} \in \mathbb{R}^{l \times l}$ and $\boldsymbol{\Psi}_{\text{shift}} \in \mathbb{R}^l$ are the affine transformation parameters. Different kinds of layers use different pairs of transform parameters $(\boldsymbol{\Psi}_{\text{scale}}, \boldsymbol{\Psi}_{\text{shift}})$, *i.e.*, different convolutional layers have their own transform, dense layers have their own, and so on.

As $\boldsymbol{\Phi}$ consists of discrete parameters which are difficult to optimize, continuous surrogates $\widehat{\boldsymbol{W}}$ are maintained for each quantized parameter $\widetilde{\boldsymbol{W}}$. $\widetilde{\boldsymbol{W}}$ is thus simply obtained by rounding the elements of $\widehat{\boldsymbol{W}}$ to the nearest integer. A straight-through estimator (Bengio et al., 2013) is used to back-propagate the gradients from the classification loss to $\widehat{\boldsymbol{W}}$. Bit-rate minimization is achieved by enforcing an entropy penalty on the surrogates $\widehat{\boldsymbol{W}}$. For a given surrogate $\widehat{\boldsymbol{W}} \in \mathbb{R}^{d \times l}$ with $d$ samples of dimension $l$, we add uniform noise $n \sim \mathcal{U} \left( -\frac{1}{2}, \frac{1}{2} \right)$, and fit $l$ probability models $\{q_j, j \in [1 \dots l]\}$ as proposed in Ballé et al. (2018). The entropy of model weights can now be minimized directly by minimizing the negative log-likelihood which serves as an approximation to the self-information $I$ (Eq. (3)). The compression term is then the sum of all the self-information terms

$$q(\widehat{\boldsymbol{W}}) = \prod_{i=1}^{C_{\text{in}} C_{\text{out}}} \prod_{j=1}^{l} q_j(\widehat{\boldsymbol{W}}_{i,j} + n_{i,j}), \quad I(\widetilde{\boldsymbol{W}}) \approx -\log_2 q(\widehat{\boldsymbol{W}}), \quad \mathcal{L}_{\text{compress}}(\boldsymbol{\Theta}) = \lambda_I \sum_{\widetilde{\phi} \in \boldsymbol{\Phi}} I(\widetilde{\phi}) \quad (3)$$

Where $\lambda_I$ is a hyper-parameter specifying relative weight of the compression loss. After training, only the quantized representations $\widetilde{\boldsymbol{W}}$ are stored by arithmetic coding using the probability tables from $q_j$. We discard the continuous surrogates post training. During inference, we load the quantized latents $\widetilde{\boldsymbol{W}}$ and decode them using the learnt decoder to obtain the continuous model weights $\boldsymbol{W}$ which is used in the model's forward pass.

## 3.2 SPARSITY PRIORS

The compression term described in the previous section encourages a smaller representation of the model in terms of the number of bits required to store the model on disk. However, once the model is decoded for inference, it can be fully dense with no reduction in terms of computation in a single forward pass of the model. To address this, we introduce a few key changes and then formulate our computation term as structured sparsity priors that lead to reduced computation. We formulate all the priors in the latent representation space to decouple from the affine transform parameters $\boldsymbol{\Psi}_{\text{scale}}, \boldsymbol{\Psi}_{\text{shift}}$ and to be consistent with the $\mathcal{L}_{\text{compress}}(\boldsymbol{\Theta})$ term that is also applied in the same space.

We observe from Eq. (2) that even if all the elements of the latent $\widetilde{\boldsymbol{W}}_i$ are zero, the resulting transformed slice $\boldsymbol{W}_i$ may still be non-zero. In order to enforce structural sparsity in the parameters in the model space $\boldsymbol{W}$, we require each $K \times K$ slice $\boldsymbol{W}_i$ to be $\boldsymbol{0}$, the zero vector. However, this is only possible if $(\boldsymbol{\Psi}_{\text{shift}} + \widetilde{\boldsymbol{W}}_i)$ is a zero vector itself or lies in the null space of $\boldsymbol{\Psi}_{\text{scale}}$. We notice that the latter does not occur in most practical situations especially when the vector $\widetilde{\boldsymbol{W}}_i$ is discrete. Therefore, we remove the shift parameter $\boldsymbol{\Psi}_{\text{shift}}$ and make the affine transform a purely linear transform.

$$\boldsymbol{W}_i = \boldsymbol{\Psi}_{\text{scale}} \widetilde{\boldsymbol{W}}_i, \quad \widetilde{\boldsymbol{W}}_i \in \mathbb{Z}^l, \boldsymbol{\Psi}_{\text{scale}} \in \mathbb{R}^{l \times l} \tag{4}$$

Note that the $j^{\text{th}}$ element in $\boldsymbol{W}_i$ is zero only if the $j^{\text{th}}$ row of $\boldsymbol{\Psi}_{\text{scale}}$ is orthogonal to $\widetilde{\boldsymbol{W}}_i$ or $\widetilde{\boldsymbol{W}}_i = \boldsymbol{0}$. The former tend to be rare or nonexistent in practice due to $\boldsymbol{\Psi}_{\text{scale}}$ being real-valued and $\widetilde{\boldsymbol{W}}_i$ being discrete. Thus, any single non-zero element in $\widetilde{\boldsymbol{W}}_i$ causes the transformed model vector $\boldsymbol{W}_i$ to be nonzero and does not yield any sparsity in the model space. Loosely, we get $\widetilde{\boldsymbol{W}}_i = \boldsymbol{0} \Leftrightarrow \boldsymbol{W}_i = \boldsymbol{0}$.

**Unstructured sparsity of latents with Gaussian prior:** Since a sparse model contains a majority of zeros, its weight distribution should peak at zero. However, the loss in Eq. (3) does not necessarily enforce zero latents, allowing for any non-zero constant value. A zero-mean Gaussian is one such distribution that enforces zero-centered latents. It also enforces other useful properties such as unimodality and symmetry, typically observed in trained uncompressed model weight distributions (Appendix B). Similar to Eq. (3), a Gaussian prior can be viewed as a compression penalty but with a Gaussian weight distribution. This corresponds to the $l_2$ norm penalty on $\widehat{\boldsymbol{W}}$.

$$q_U(\widehat{\boldsymbol{W}}) = \prod_{i=1}^{C_{\text{in}}C_{\text{out}}} \prod_{j=1}^{l} \frac{1}{\sqrt{2\pi\sigma^2}} e^{-\frac{1}{2}\widehat{\boldsymbol{W}}_{i,j}^2} \quad \text{and} \quad I_U(\widetilde{\boldsymbol{W}}) \approx -\log_2 q_U(\widehat{\boldsymbol{W}}) \tag{5}$$

$$\mathcal{L}_{\text{gaussian}}(\boldsymbol{\Theta}) = \lambda_U \sum_{\widetilde{\boldsymbol{W}} \in \boldsymbol{\Phi}} I_U(\widetilde{\boldsymbol{W}}) = \lambda_U \sum_{\widetilde{\boldsymbol{W}} \in \boldsymbol{\Phi}} \sum_{i=1}^{C_{\text{in}}C_{\text{out}}} \sum_{j=1}^{l} \|\widehat{\boldsymbol{W}}_{i,j}\|_2 \tag{6}$$

Where $\lambda_U$ is the tradeoff parameter controlling how closely the weights follow a Gaussian distribution compared to the the fully factorized distribution from the probability models in Eq. (3). A laplacian prior, resulting in an $l_1$ penalty, can also be used but the difference between the two distributions is negligible due to the effects of the quantization on $\widehat{\boldsymbol{W}}$. We experimented with both priors and observed similar performance as expected (refer Appendix G). Through experiments in Sec. 4, we show that this prior not only enforces model sparsity by design but also improves model compression.

**Structured sparsity of latents with group lasso:** The Gaussian prior encourages individual weight values in the latents to be close to zero. However, from Eq. (4), we see that entire slices $\widetilde{\boldsymbol{W}}_i$ should be zero vectors to improve sparsity in the model space. Note that each slice $\widetilde{\boldsymbol{W}}_i$ can be represented as a group belonging to $\widetilde{\boldsymbol{W}}$, the set of groups (or slices). Thus, to enforce individual slices to go to zero as a whole, we propose to use a group sparsity regularization (Yuan & Lin, 2006) on each $\widehat{\boldsymbol{W}}_i$ slice as follows.

$$\mathcal{L}_{\text{group}}(\boldsymbol{\Theta}) = \lambda_S \sum_{\widetilde{\boldsymbol{W}} \in \boldsymbol{\Phi}} \sum_{i=1}^{C_{\text{in}} \times C_{\text{out}}} \sqrt{\rho} \|\widehat{\boldsymbol{W}}_i\|_2 \tag{7}$$

where $\rho$ accounts for varying group sizes and $\lambda_S$ is the tradeoff parameter for the structured sparsity.

### 3.3 JOINT OPTIMIZATION OBJECTIVE

The overall loss function is the combination of cross-entropy loss (for classification), self-information, and regularization for structured and unstructured sparsity of the latents as follows:

$$\underbrace{\overbrace{\sum_{(\mathbf{x},y) \sim D} -\log p(y|\mathbf{x}; \widetilde{\boldsymbol{W}})}^{\text{Accuracy}}}_{\text{Cross Entropy}} + \underbrace{\overbrace{\lambda_I \sum_{\widetilde{\boldsymbol{\phi}} \in \boldsymbol{\Phi}} I(\widetilde{\boldsymbol{\phi}})}^{\text{Disk Size}}}_{\substack{\text{Parameter} \\ \text{Entropy}}} + \overbrace{\underbrace{\lambda_U \sum_{\widetilde{\boldsymbol{W}} \in \boldsymbol{\Phi}} \sum_{i=1}^{C_{\text{in}}C_{\text{out}}} \sum_{j=1}^{l} \|\widehat{\boldsymbol{W}}_{i,j}\|_2}_{\substack{\text{Unstructured} \\ \text{Sparsity}}} + \underbrace{\lambda_S \sum_{\widetilde{\boldsymbol{W}} \in \boldsymbol{\Phi}} \sum_{i=1}^{C_{\text{in}} \times C_{\text{out}}} \sqrt{\rho} \|\widehat{\boldsymbol{W}}_i\|_2}_{\substack{\text{Structured} \\ \text{Sparsity}}}}^{\text{Computation}}$$

$$\tag{8}$$

The above objective is fully differentiable and can be minimized end-to-end by any gradient-based optimizer. Note that while there are three tradeoff parameters, each of them is relatively independent and intuitively controls different aspects of the network. $\lambda_I$ controls for model size, $\lambda_U$ for enforcing a zero-mean prior necessary for sparsity, and $\lambda_S$ for slice sparsity and hence computation during network inference. $l_2$ weight decay on model weights is typically used as a regularization for improving generalization. However, our motivation for a Gaussian prior is for the objective detailed in Sec. 3.2. Furthermore, our group lasso prior and the Gaussian prior are applied in the latent representation space rather than the model space as in Wen et al. (2016). Due to the presence of quantization and decoding, the effect of these priors on training is significantly different than when applying these priors directly to the model weights. We analyze the effect of these priors in Sec. 4.

### 3.4 IMPLEMENTATION DETAILS

**Datasets.** We consider three datasets in our experiments. CIFAR-10 and CIFAR-100 datasets (Krizhevsky et al., 2009) consist of 50000 training and 10000 test color images each of size $32 \times 32$. For large scale experiments, we use ILSVRC2012 (ImageNet) dataset (Deng et al., 2009). It has 1.2 million images for training, 50000 images for the test and 1000 classes.

**Network Architectures.** For CIFAR-10/100 datasets, we show results using - VGG-16 (Simonyan & Zisserman, 2014) and ResNet-20 with a width multiplier of 4 (ResNet-20-4) (He et al., 2016). VGG-16 is a commonly used architecture consisting of 13 convolutional layers of kernel size $3 \times 3$ and 3 dense or fully-connected layers. Dense layers are resized to adapt to CIFAR's $32 \times 32$ image size, as done in baseline approaches. ResNet-20-4 consists of 3 ResNet groups, each with 3 residual blocks. All convolutional layers are of size $3 \times 3$, along with a final dense layer. For the ImageNet experiments, we use ResNet-18/50 networks with one $7 \times 7$ convolutional layer, multiple 3 and $1 \times 1$ convolutional layers, and the final dense layer. We also run experiments with MobileNet-V2 (Sandler et al., 2018), containing depthwise separable convolutions and inverted Bottleneck blocks.

We use the Adam optimizer (Kingma & Ba, 2014) for updating all parameters of our models. The entropy model parameters are optimized with a learning rate of $10^{-4}$ for all our experiments. The remaining parameters are optimized with a learning rate of $0.01$ for CIFAR-10 experiments and a learning rate of $0.02$ for ResNet-18/50 on ImageNet with a cyclic schedule. Our model compression results are reported using the torchac library (Mentzer et al., 2019) which does arithmetic coding of the weights given probability tables for the quantized values which we obtain from the probability models. We do not compress the biases and batch normalization (BN) parameters and include the additional sizes from these parameters as well as the parameters $\mathbf{\Psi}_{\text{scale}}$ when reporting model size. Parameter groups for different networks based on types of convolutional/dense layers are provided in the appendix (Appendix E).

## 4 ANALYSIS

Recall from Eq. (8), we proposed two sparsity coefficients: $\lambda_U$ for unstructured sparsity, and $\lambda_S$ for structured sparsity. Note that the sparsity terms not only improve the model's inference speed (by increasing slice sparsity and reducing the number of FLOPs) but also reduce the entropy of latent weights $\widetilde{\mathbf{W}}$ as most of the weights become zero after quantization. By varying the two sparsity coefficients, one for each sparsity term, we obtain different points on the Pareto curves for

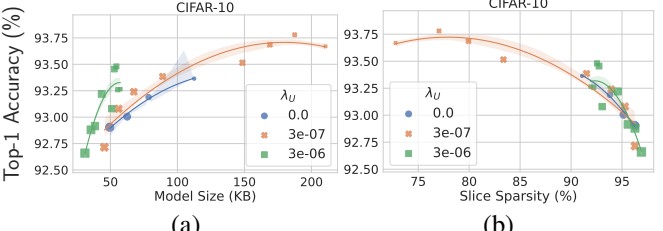

(a)                       (b)

Figure 3: **Effect of unstructured sparsity coefficient $\lambda_U$.** We show plots for 3 values of $\lambda_U$ varying $\lambda_S$ to obtain the Pareto curves on CIFAR-10. Higher $\lambda_U$ improves accuracy-model size tradeoff (left) by a good margin and also the accuracy-slice sparsity (right) by a small amount. Shaded areas represent the confidence intervals of the regression fit. Best viewed in color.

accuracy *vs*. model size trade-off and accuracy *vs*. slice sparsity trade-off. In this section, we study each of these trade-offs extensively. We use slice sparsity as a proxy for computational complexity

or inference speed in this section for brevity, and revisit the computational complexity in terms of actual wall-clock inference time in Sec. 5.2. We use a constant compression coefficient $\lambda_I = 10^{-4}$. We show Pareto curves of model performance (accuracy) *vs.* model size (bit-rate), and the slice sparsity (%) by keeping one of $\lambda_U, \lambda_S$ fixed and varying the other. We analyze model sizes for the compressed parameters in this section as the remaining parameter sizes are constant. All results in this section are obtained by averaging over 3 runs with varying random seeds.

## 4.1    Effect of Unstructured Sparsity Regularization

Fig. 3 shows model performance for three different values of $\lambda_U$, while varying $\lambda_S$ for obtaining the Pareto trade-off curve for each case. Fig. 3(a) and 3(b) show the impact on the top-1 accuracy as a function of model size and slice sparsity respectively. We see that increasing $\lambda_U$ (blue circles to orange crosses to green squares) improves the accuracy *vs.* model size tradeoff curve by a fair margin while also slightly improving the accuracy *vs.* slice sparsity curve. This is to be expected as higher $\lambda_U$ leads to a higher number of zeros in the latent representations leading to lesser entropy and consequently lower model size. We also see a marginal gain in slice sparsity even though the Gaussian prior doesn't directly optimize for it as a higher number of zeros eventually leads to more slices entirely going to zero. Thus, we see that the Gaussian prior helps with improving model size while also marginally improving slice sparsity.

## 4.2    Effect of Structured Sparsity Regularization

Fig. 4 shows model performance for three values of $\lambda_s$, while varying $\lambda_U$ to obtain the Pareto curves. Again, Fig. 4(a) and 4(b) show the impact on the top-1 accuracy as a function of model size and slice sparsity respectively. The highest value of $\lambda_S$ (green squares) shows a marked improvement of the accuracy-sparsity curve in Fig. 4(b). This shows that the group lasso regularization is effective in improving the slice sparsity of the model necessary for computational benefits. Additionally, we see a small improvement in the accuracy-model size curve as higher $\lambda_S$ promotes a higher number of zero slices which indirectly leads to lower entropy of the overall latent representations.

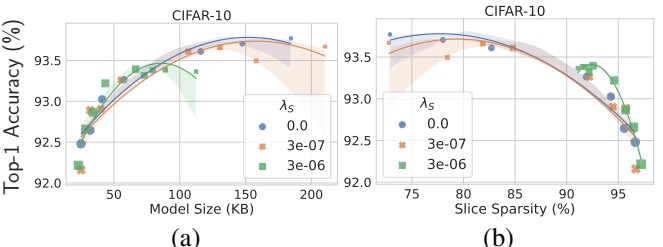

Figure 4: **Effect of structured sparsity coefficient $\lambda_S$.** We show plots for 3 values of $\lambda_S$ varying $\lambda_U$ to obtain the Pareto curves. Fig. (a) and (b) correspond to Accuracy-Model Size trade-off and Accuracy-Slice Sparsity trade-off respectively on CIFAR-10. Highest $\lambda_S$ improves the accuracy-model sparsity tradeoff curve showing the importance of group lasso regularization for improving slice sparsity. Shaded areas represent the confidence intervals of the regression fit. Best viewed in color.

## 4.3    Structured *vs.* unstructured sparsity regularization

While both structured and unstructured sparsity regularization help improve slice sparsity, the latter optimizes for it indirectly. Fig. 5 shows the effect of $\lambda_U, \lambda_S$ on unstructured and slice sparsity of the latent representations. As expected, we observe that increasing $\lambda_U$ (corresponding to larger points), for a fixed $\lambda_S$, increases the sparsity of the latent representations while indirectly increasing slice sparsity as well. However, as we increase $\lambda_S$ we notice that the plots shift upwards implying higher structural sparsity for any fixed value of unstructured sparsity. Therefore, we see that our structured sparsity prior is effective in forcing non zero weights to lie in fewer weight slices thus leading to higher structured sparsity, ultimately leading to speedups in model inference. Note that while the structured sparsity constraint directly optimizes for the slice sparsity, the unstructured sparsity promotes sparsity within a group/slice, akin to Simon et al. (2013). Unstructured sparsity has a strong effect on model size due to a large number of zeros and lower entropy as we show in Sec.4.1 while structured sparsity constraint directly affects slice sparsity as shown above and in Sec.4.2.

Table 1: Comparison of our approach against other model compression techniques. We show two cases of our method: Best and Extreme. Best corresponds to our best model in terms of error rate and compression factor while Extreme is matching the range of error of baselines if exists. We achieve higher compression along with the added computational benefits of high slice sparsity. "−" implies that the work performs pruning but does not report numbers in the paper.

| Algorithm | Size (KB) | Error (Top-1 %) | Sparsity (%) |
|---|---|---|---|
| VGG-16 (CIFAR-10) | | | |
| Uncompressed | 60 MB (1×) | 6.6 | 0.0 |
| Louizos et al. (2017) | 525 (116×) | 9.2 | 94.5 |
| Wiedemann et al. (2019) | 960 (62×) | 9.0 | 92.4 |
| Havasi et al. (2018) | 168 (452×) | 10.0 | 0.0 |
| Oktay et al. (2020) | 101 (590×) | 10.0 | 0.0 |
| LilNetX (Best) | 129 (465x) | 7.4 | 97.4 |
| LilNetX (Extreme) | **76 (800x)** | 10.0 | **99.2** |
| ResNet-20-4 (CIFAR-10) | | | |
| Uncompressed | 17.2 MB (1×) | 5.5 | 0.0 |
| Oktay et al. (2020) | 128 (134×) | 8.8 | 0.0 |
| LilNetX (Best) | 139 (123×) | 6.0 | 88.5 |
| LilNetX (Extreme) | **66 (282×)** | 8.5 | **97.9** |
| ResNet-20-4 (CIFAR-100) | | | |
| Uncompressed | 17.2 MB (1×) | 26.8 | 0.0 |
| LilNetX (Best) | 125 (137×) | 27.0 | 86.6 |
| LilNetX (Extreme) | **76 (226×)** | 30.9 | **97.1** |

| Algorithm | Size (MB) | Error (Top-1 %) | Sparsity (%) |
|---|---|---|---|
| ResNet-18 (ImageNet) | | | |
| Uncompressed | 46.70 (1×) | 30.0 | 0.0 |
| Dubey et al. (2018) | 3.11 (15×) | 32.0 | - |
| Young et al. (2021) | 2.78 (17×) | 31.8 | 0.0 |
| Oktay et al. (2020) | 1.97 (24×) | 30.0 | 0.0 |
| LilNetX (Best) | 1.58 (30×) | 30.1 | 33.3 |
| LilNetX (Extreme) | **0.86 (54×)** | 32.3 | **65.2** |
| ResNet-50 (ImageNet) | | | |
| Uncompressed | 102.00 (1×) | 23.7 | 0.0 |
| Lin et al. (2019) | 39.91 (3×) | 27.7 | 39.2 |
| Frankle et al. (2019) | 20.40 (5×) | 24.0 | 49.1 |
| Dubey et al. (2018) | 6.46 (16×) | 26.0 | - |
| Wiedemann et al. (2019) | 6.06 (17×) | 25.9 | 25.4 |
| Oktay et al. (2020) | 5.49 (19×) | 26.0 | 0.0 |
| Young et al. (2021) | 4.39 (21×) | 25.8 | 0.0 |
| LilNetX (Best) | 3.96 (26×) | 25.4 | 66.7 |
| LilNetX (Extreme) | **2.96 (34×)** | 25.8 | **81.7** |
| MobileNet-V2 (ImageNet) | | | |
| Uncompressed | 14.00 (1×) | 32.7 | 0.0 |
| Tu et al. (2020) | 10.30 (1×) | 32.7 | 25.0 |
| Wang et al. (2019) | 0.95 (15×) | 33.5 | 0.0 |
| LilNetX (Best) | 0.67 (21×) | 32.8 | 56.8 |
| LilNetX (Extreme) | **0.56 (25×)** | 33.3 | **64.7** |

# 5 EXPERIMENTS

All our models are trained from scratch. CIFAR-10/100 experiments are trained for 200 epochs. We use the FFCV library (Leclerc et al., 2022) for faster ImageNet training, with a batch size of 512 for ResNet-18/50 split across 4 GPUs. We train ResNet-18/50 for 35 epochs to keep the range of the uncompressed network accuracies similar to other works for a fair comparison. We show strong performance in terms of model compression and sparsity outperforming existing model compression works while converging faster with relatively fewer epochs.

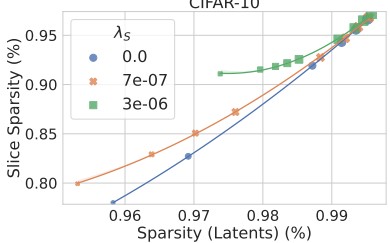

Figure 5: **Structured vs. unstructured sparsity regularization.** Higher $\lambda_S$ improves slice sparsity (y-axis) for a given level of unstructured sparsity (x-axis).

## 5.1 COMPARISON WITH COMPRESSION METHODS

As discussed in Sec. 2, existing approaches for model compression follow either **quantization**, **pruning**, or both. We compare with the state of the art methods in each of these two categories. Among model quantization methods, we use Oktay et al. (2020) and Young et al. (2021) for comparison, with the latter offering speedups via mix-precision inference albeit on specialized hardware. Our results are summarized in Table 1. Unless otherwise noted, we use the numbers reported by the original papers. Since we do not have access to many prior art models, we compare using slice sparsity. For the CIFAR-10 dataset, we achieve the best results in compression while also achieving a lower Top-1 error rate for both VGG-16 and ResNet-20-4. For VGG-16 we obtain the best performance in the error range of $t7\%$ at $129KB$ which is a 465x compression compared to the baseline model. At the ∼10% error range, we outperform Oktay et al. (2020) in terms of model compression and also with a 99.2% slice sparsity. For ResNet-20-4, compared to Oktay et al. (2020), we achieve almost twice the compression rate at a similar error rate of ∼8.5%, simultaneously achieving extremely high levels of slice sparsity (97.9%). Similar results hold for the case of CIFAR-100 where we achieve a $137\times$ compression in model size with 86.7% slice sparsity and little to no drop in accuracy compared to the uncompressed model.

For ResNet-18 trained on ImageNet, we achieve $30\times$ compression as compared to the uncompressed model with almost equal error rate outperforming Oktay et al. (2020). The compressed network achieves a sparsity of 33.3%. For an extreme compression case, we achieve higher levels of com-

pression ($54\times$) and sparsity ($\sim 65\%$) at the cost of $\sim 2\%$ accuracy compared to uncompressed model. For ResNet-50, our best model achieves a compression rate of $26\times$, along with $66.7\%$ slice sparsity. An extreme case of our model achieves a higher compression rate of $34\times$ with sparsity of $81.7\%$ compared to the next best work of Young et al. (2021) with a rate of $21\times$ at a similar error rate. We provide additional baseline comparisons for ResNet-50 in the supplementary material.

Finally, for MobileNet-V2 which is already lightweight and optimized for computational efficiency, we still achieve $21\times$ compression compared to the dense model along with $56.8\%$ slice sparsity with almost no drop in accuracy. This is especially beneficial for MobileNets which consist of Depthwise Separable Convolutions as removing entire 2D slices of the convolutional weight allows for directly removing the input activation map's corresponding channels leading to lower FLOPs. We outperform other baselines at a similar accuracy both in terms of model compression and sparsity.

We conclude that our framework outperforms state-of-the-art (SOTA) approaches in model compression by a fair margin while achieving network weights sparsification for computational gains.

## 5.2 Utilizing Slice Sparsity for Inference Speedups

While we show compression gains with respect to the SOTA in Sec. 5.1, here we highlight the computational gains we get through slice sparsity. We measure the speedups of the trained sparse models by obtaining the ratio of the throughput (images/second) of the sparse compressed models to that of the dense uncompressed models. Speedups are measured on a single core of an AMD EPYC

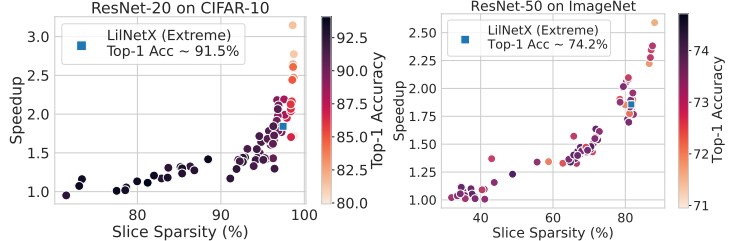

Figure 6: **Speedups _vs_. Slice Sparsity (%).** Speedup is the ratio of CPU throughput for the sparse models to the dense models. For both cases of ResNet-20 on CIFAR-10 (left) and ResNet-50 on ImageNet (right), we obtain practical inference speedups utilizing the sparsity of the compressed models.

7302 16-Core Processor with a batch size of 16. We show inference results for two cases: ResNet-20 on CIFAR-10 and ResNet-50 on ImageNet. As a high slice sparsity provides the added benefits of full structured sparsity (when all slices in a filter/channel are zero), we show speedups for ResNet-20 by utilizing fully structured sparsity, removing entire filters or input channels of weight tensors which are all zeros. Niu et al. (2020) show that slice sparsity, along with pattern sparsity, provide inference speedups on mobile devices via compiler-assisted optimizations. However, due to the lack of an open-source code base, we utilize the DeepSparse engine (Kurtz et al., 2020) which also exploits this sparsity for CPU inference speedups. Results are shown in Fig. 6. For CIFAR-10 (left), even when only exploiting fully structured sparsity, we achieve nearly $2\times$ levels of speedups for $95\%$ slice sparsity. Speedups scale almost exponentially for sparsity$>95\%$. For ImageNet (right), we obtain $1.86\times$ speedup compared to the dense uncompressed model at $81.7\%$ slice sparsity and even faster inference times for sparsity$>85\%$. Therefore, our framework offers practical inference speedups via slice sparsity with no hardware modifications, along with high levels of model compression.

## 6 Conclusion

We propose a novel framework for training a deep neural network, while simultaneously optimizing the model size to reduce storage cost, and structured sparsity, to reduce computation cost. To the best of our knowledge, this is the first work on model compression that add priors for structured pruning of weights in quantized latent representation space.

Experiments on three datasets and three network architectures show that our approach achieves state-of-the-art performance in terms of simultaneous compression and reduction in computation which directly translate to inference speedups. We also perform extensive ablation studies to verify that the proposed sparsity priors allow us to easily control the accuracy-compression-computation trade-off, which is an important consideration for the practical deployment of models.

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

APPENDIX

## A    ADDITIONAL BASELINE COMPARISON

We compare our approach with additional pruning and quantization approaches in Table 2 for ResNet-50 trained on ImageNet. We see that we continue to achieve high levels of model compression along with slice sparsity for inference speedups. Yuan et al. (2020) achieve high levels of sparsity but are unstructured requiring dedicated hardware to obtain speedups. A similar case holds for the quantization approaches of Zhao et al. (2019); Jain et al. (2020) which can obtain inference speedups but with hardware optimized for 4-bit and 8-bit integer arithmetic. Additionally, they typically require post-hoc training stages (Jain et al., 2020) to improve performance after quantization while our approach is a single stage trained end-to-end.

Table 2: Comparison of our approach with other pruning and quantization approaches for ResNet-50 trained on ImageNet. We continue to achieve the most compression along with high slice sparsity. * denotes that the sparsity is unstructured and do not directly translate to computational benefits.

| Algorithm | Size (MB) | Error (Top-1 %) | Sparsity (%) |
|---|---|---|---|
| ResNet-50 (ImageNet) | | | |
| Uncompressed | 102.00 (1×) | 23.7 | 0.0 |
| Savarese et al. (2020) | 8.36 (12×) | 24.5 | 91.8* |
| Yuan et al. (2020) | 38.76 (3×) | 24.8 | 38.8 |
| Zhao et al. (2019) (4 bit) | 12.75 (8×) | 33.8 | 0.0 |
| Zhao et al. (2019) (8 bit) | 25.5 (4×) | 25.3 | 0.0 |
| Jain et al. (2020) (4 bit w/ retraining) | 12.75 (8×) | 25.6 | 0.0 |
| Jain et al. (2020) (8 bit w/o retraining) | 25.5 (4×) | 25.7 | 0.0 |
| Jain et al. (2020) (8 bit w/ retraining) | 25.5 (4×) | 24.6 | 0.0 |
| LilNetX (Best) | 3.96 (26×) | 25.4 | 66.7 |
| LilNetX (Extreme) | **2.96 (34×)** | 25.8 | **81.7** |

## B    HISTOGRAM OF WEIGHTS FOR DENSE UNCOMPRESSED MODEL

We obtain the histogram of weights of the various types of layers of a dense uncompressed ResNet-50 model trained on ImageNet with only the cross entropy loss. We do not apply any weight decay in order to avoid enforcing any distribution on the weights. Results are shown in Fig. 7. We show histograms for $1 \times 1$, $3 \times 3$, $7 \times 7$ convolutions as well as for the dense layer. For $3 \times 3$ and $7 \times 7$ convolutions, we pick a random dimension from a 9-dimensional or a 49-dimensional slice respectively, to highlight the histograms, as a single probability model is fit to each dimension as shown in Eq. (3). We see that the distributions naturally follow unimodality and are more or less zero-centered even without any weight decay regularization. The $7 \times 7$ convolution weight distribution is less continuous due to relatively fewer weight values per dimension (192) but still weakly exhibits the property of unimodality and symmetry. This shows that networks trained with vanilla cross entropy loss prefer such distributions naturally. However, the probability models in Eq. (3) do not enforce any such distribution and can take on any random distribution. Thus, enforcing a Gaussian prior as proposed in Sec. 3.2 promotes unimodality and symmetry of the weight distributions which can be beneficial for network performance.

## C    HISTOGRAM OF WEIGHTS FOR QUANTIZED LATENTS

To provide insights into the effect of our quantization, we visualize the histogram of the quantized latents as well for different weight groups. Results are shown in Fig. 8. We see that we obtain high levels of 0s on almost all weight groups spanning different types of convolutional layers as well as the final dense layer. Fewer number of zeros are present in the initial $7 \times 7$ convolution similar to the uncompressed weights as shown in Fig. 7 highlighting its importance in the network. Additionally, high amount of elements are zeros in $3 \times 3$ convolutions highlighting their redundancy and potential for compression compared to other convolutional layers or the dense layer.

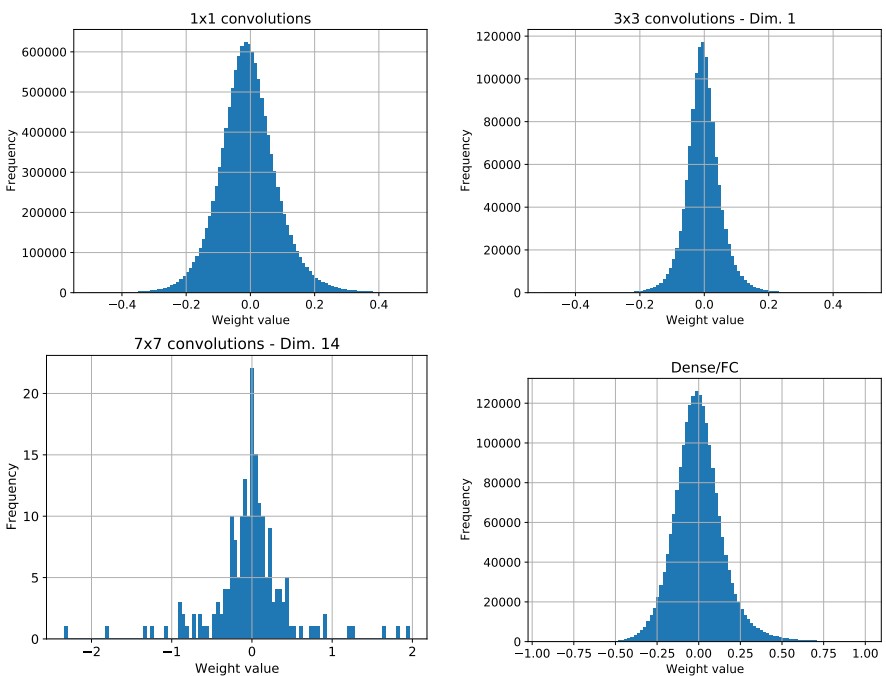

Figure 7: Histogram of the weights for various types of layers of a ResNet-50 model trained on ImageNet without weight decay. For $3 \times 3$ or $7 \times 7$ convolutions we pick a random dimension of a 9-dimensional or 49-dimensional slice to highlight the weight distribution for each dimension. Note the unimodal and zero-centered nature of each of these distributions, even without enforcing weight decay, highlighting the importance of the Gaussian prior as proposed in Sec. 3.2

## D   COMPARISON OF MODEL COMPRESSION WITH ENTROPY CODING AND SPARSE MATRIX FORMATS

Instead of entropy coding, the sparse matrices can additionally be compressed using sparse matrix formats. We choose the two popular formats of Compressed Sparse Row (CSR) or Coordinate Format(COO). Results are summarized in Table 3 for our best run for ResNet-50 shown in Table 1 in the main paper. We see that entropy coding far outperforms the sparse formats of CSR and COO with COO obtaining better compression rates than CSR. This is expected as CSR/COO achieves high levels of compression only with extremely high levels of sparsity. With an unstructured sparsity level of ~80%, storing only the non zero weights itself (and not their indices) provides a maximum compression of $5\times$.

Table 3: **Sparse formats:** Comparison of the effect of entropy coding *vs.* sparse matrix formats of CSR, COO on model compression of a ResNet-50 trained on ImageNet. We show the model size in MB of the latent weights along with the sparsity of the model weights.

| Entropy Coding | CSR | COO | Slice Sparsity (%) | Unstructured Sparsity (%) |
|---|---|---|---|---|
| **3.96 (26×)** | 57 (2×) | 30 (3×) | 66.7 | 78.8 |

## E   PARAMETER GROUPS FOR VARIOUS NETWORKS

We share weight decoders and probability models for different parameter groups of a network which can be seen as being drawn from similar weight distributions. This limits the overhead in storing the weights of the corresponding decoders. We list the types of parameter groups for each network as follows:

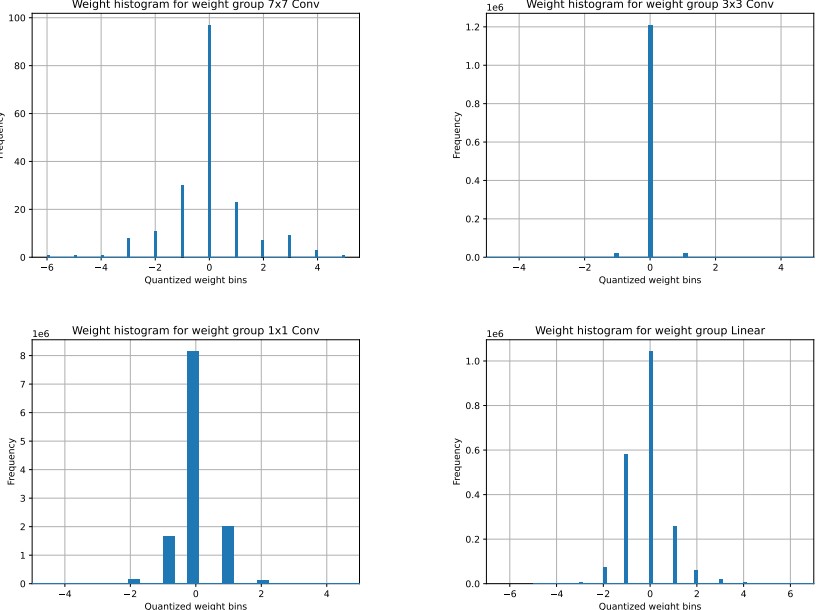

Figure 8: Histogram of the quantized latents for various types of layers of a ResNet-50 model trained on ImageNet. For $3 \times 3$ or $7 \times 7$ convolutions we pick a random dimension of a 9-dimensional or 49-dimensional slice to highlight the weight distribution for each dimension. Note the high level of 0s obtained due to our sparsity priors.

- VGG-16 consists of a parameter group for each dense layer and a parameter group for all $3 \times 3$ convolutions leading to four weight decoders/probability models for each parameter group.
- For ResNet-20-4 we use zero padding shortcut type A as defined in He et al. (2016), which leads to only 2 parameter groups, one for the final dense layer and the other for all $3 \times 3$ convolutions.
- For ResNet-18 trained on ImageNet, we use three parameter groups, for the initial 7x7 convolution, $3 \times 3$ convolutions, as well as the dense layer.
- ResNet-50 consists of an additional parameter group for $1 \times 1$ convolutions compared to ResNet-18.
- MobileNet-V2 consists of 3 parameter groups for the initial 3x3 convolution, final dense layer and the remaining 3x3 convolution.

## F  STANDARD ERROR FOR MULTIPLE RUNS

Sec. 5 in the main paper shows results when averaged across 3 seeds. In this section, we additionally provide the standard errors across the 3 random seeds. Results are summarized in Fig. 9 for the two datasets of CIFAR-10/100. CIFAR-10 shows little to no standard error both in the x-axis (model size) and y-axis (top-1 validation accuracy). This suggests that the training is stable for different random seeds. For CIFAR-100 however, we observe large error in the top-1 validation accuracy. We attribute this to the slow convergence for CIFAR-100 also highlighted in Fig. 10.

**CIFAR-100 Convergence**:We analyze the convergence of 3 different runs for ResNet-20-4 trained on the CIFAR-100 dataset with varying values of $\lambda_S$ and $\lambda_U$. Results are shown in Fig. 10 when trained for 200 epochs. We see that validation accuracy (on the right y-axis) continues to increase towards the end of training between 190-200 epochs. At the same time, validation loss (on the left y-axis) also decreases. This suggests that the model hasn't fully converged by the end of 200 epochs. We hypothesize that this is an artifact of the dataset as well as the cosine decay schedule

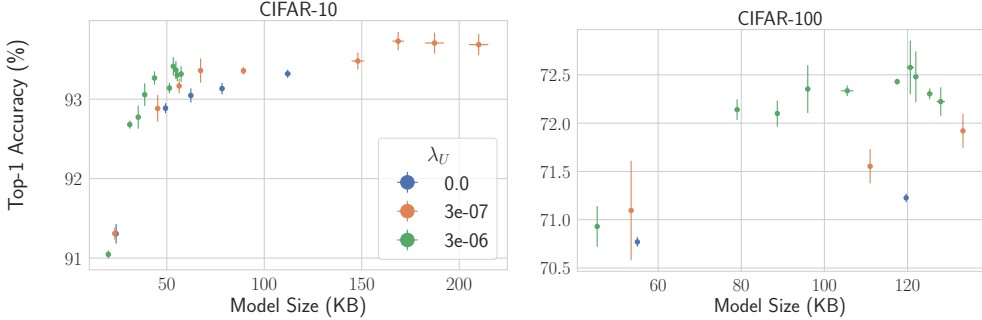

Figure 9: Scatter plots with horizontal and vertical error bars for ResNet-20-4 trained on CIFAR-10/100. For a different random seed, model size changes leading to the error bar in the x-axis while the vertical bar represents the top-1 validation accuracy error on the y-axis. There is very little variance in CIFAR-10 and slightly higher for CIFAR-100 due to slow convergence as shown in Fig. 10

where learning rate decreases drastically towards the end of training and is not maintained for longer for better convergence.

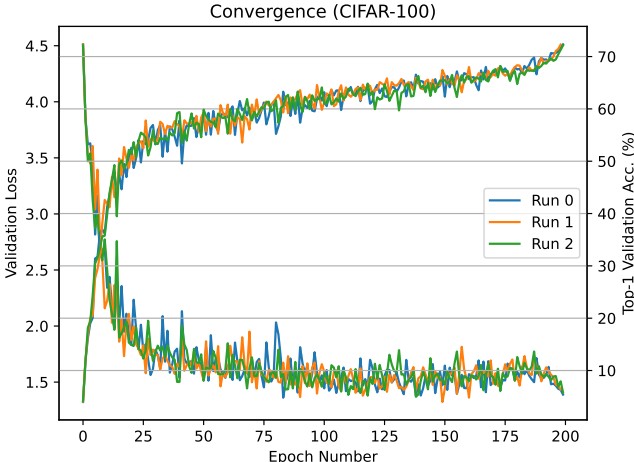

Figure 10: Convergence plots for 3 ResNet-20-4 runs on CIFAR-100. We see that loss (left axis) as well as top-1 validation accuracy (right axis) do not stabilize towards the end of training and respectively decrease/increase sharply suggesting that the training has not fully converged.

# G  $l_2$ *vs.* $l_1$ *vs.* $l_\infty$ NORM

In this section, we analyze the effect of different types of norm for both individual weights and groups. For individual weights, we compare the $l_2$ norm with the $l_1$ norm while for the group norm, we compare the $l_2$ norm with the $l_\infty$ norm ($l_1$ weight norm is same as $l_1$ group norm due to sum of absolutes). Results are summarized in Fig. 11 where top/bottom rows are for CIFAR-10/100 respectively. We see that $l_2$ group norm outperforms its $l_\infty$ counterpart for both datasets. However, $l_1$ norm has little additional effect in terms of $l_2$ weight norm. Additionally, the $l_2$ group norm yields lesser slice sparsity for a given sparsity (c,g) highlighting the importance of $l_\infty$ for high structured sparsity. While $l_\infty$ leads to higher sparsity, it also shows higher model size for a given slice sparsity. Thus, there is an inherent tradeoff for $l_\infty$ which leads to more sparsity but also larger model sizes (d,h).

# H  INITIALIZATION OF CONTINUOUS SURROGATES

The initialization of the continuous surrogate $\widehat{W}$ of a latent space weight $\widetilde{W}$ and the decoder matrix $\Psi$ plays an important in the neural network training. Naïve He initialization (He et al., 2015) commonly used in training ResNet classifiers does not work in our case since small values of $\widehat{W}$ get rounded to zero before decoding. Such an initialization results in zero gradients for updating the parameters and the loss becomes stagnant. To overcome this issue, we propose a modification to the initialization of the different parameters. In our framework, we recap that the decoded weights used in a forward pass are obtained using

$$W = \mathrm{reshape}(\widetilde{W}\Psi) \tag{9}$$

where $\widetilde{W}$ is a matrix in $\mathbb{Z}^{C_{\mathrm{in}}C_{\mathrm{out}}\times l}$ and $\Psi$ is a matrix in $\mathbb{Z}^{l\times l}$ (where $l = 1$ for dense weights (and biases) while $l = K^2$ for convolutional weights).

Our goal is to initialize $\widehat{W}$ and $\Psi$ such that the decoded weights $W$ follow He initialization. First, since $\widehat{W}$ is rounded to nearest integer (to obtain latent space weights $\widetilde{W}$), we assume its elements to be drawn from a uniform distribution in $[-b, b]$ where $b > 0.5$ in order to enforce atleast some non-zero weights after rounding to nearest integer. Next, we take the elements of $\Psi$ to be a normal distribution with mean 0 and variance $v$.

Assuming the parameters to be i.i.d., and $\mathrm{Var}(X)$ denoting the variance of any individual element in matrix $X$,

$$\mathrm{Var}(W) = l \times \mathrm{Var}(\Psi) \times \mathrm{Var}(\widetilde{W}) \tag{10}$$

Assuming a RELU activation, with $f$ denoting the total number of channels (fan-in or fan-out) for a layer, LHS of Eq. (10), using the He initializer becomes $\frac{2}{f}$, RHS on the other hand can be obtained analytically

$$\frac{2}{f} = l \times v \times \frac{(2b+1)^2 - 1}{12}$$

$$\implies b = \frac{\sqrt{\frac{24}{lvf} + 1} - 1}{2}, v = \frac{24}{lf\left((2b+1)^2 - 1\right)} \tag{11}$$

Eq. (11) gives us a relationship between $b$ (defining the uniform distribution of $\widehat{W}$) and $v$ (defining the normal distribution of $\Psi$). Note that $l$ and $f$ values are constant and known for each layer.

For a weight decoder corresponding to a parameter group, the maximum value of $f$ in that group enforces the smallest value of $b$ which should be above a minimum limit $b_{\min}$. Denoting $f_{\max}$ as the maximum fan-in or fan-out value for a parameter group, we get

$$v = \frac{24}{lf_{\max}\left((2b_{\min}+1)^2 - 1\right)}$$

$$\implies b = \frac{\sqrt{\frac{f_{\max}}{f}\left((2b_{\min}+1)^2 - 1\right) + 1} - 1}{2} \tag{12}$$

The hyperparameter $b_{\min}$ then refers to the minimum boundary any latent space parameter can take in the network. By calculating the values of $v$ based on $f_{\max}, b_{\min}$ and $b$ for various parameters based on the corresponding value of $f$, we then initialize the elements of $\widehat{W}$ to be drawn from a uniform distribution in the interval $[-b, b]$ and elements of $\Psi$ to be drawn from $\mathcal{N}(0, v)$.

Note that $f = f_{\max} \implies b = b_{\min}$ which shows that the minimum boundary corresponds to the layer with maximum channels (fan-in or fan-out) $f$.

By choosing an appropriate value of $b_{\min}$ we obtain good initial values of the gradient which allows the network to converge well as training progresses. $b_{\min}$ offers an intuitive way of initializing the discrete weights. Too small a value leads to most of the weights being set to zero while too large a value can lead to exploding gradients. In practice, we find that this initialization approach works well for Cifar experiments. For ImageNet experiments, we assume a normal distribution instead of uniform distribution for $\widehat{W}$ with a sufficiently high variance for the network to train.

## I  LICENSE

Table 4: **Licenses of datasets**.

| Dataset | License |
|---|---|
| CIFAR-10 Krizhevsky et al. (2009) | MIT |
| CIFAR-100 Krizhevsky et al. (2009) | MIT |
| ImageNet Deng et al. (2009) | BSD 3-Clause |

Table 4 lists all datasets we used and their licenses.

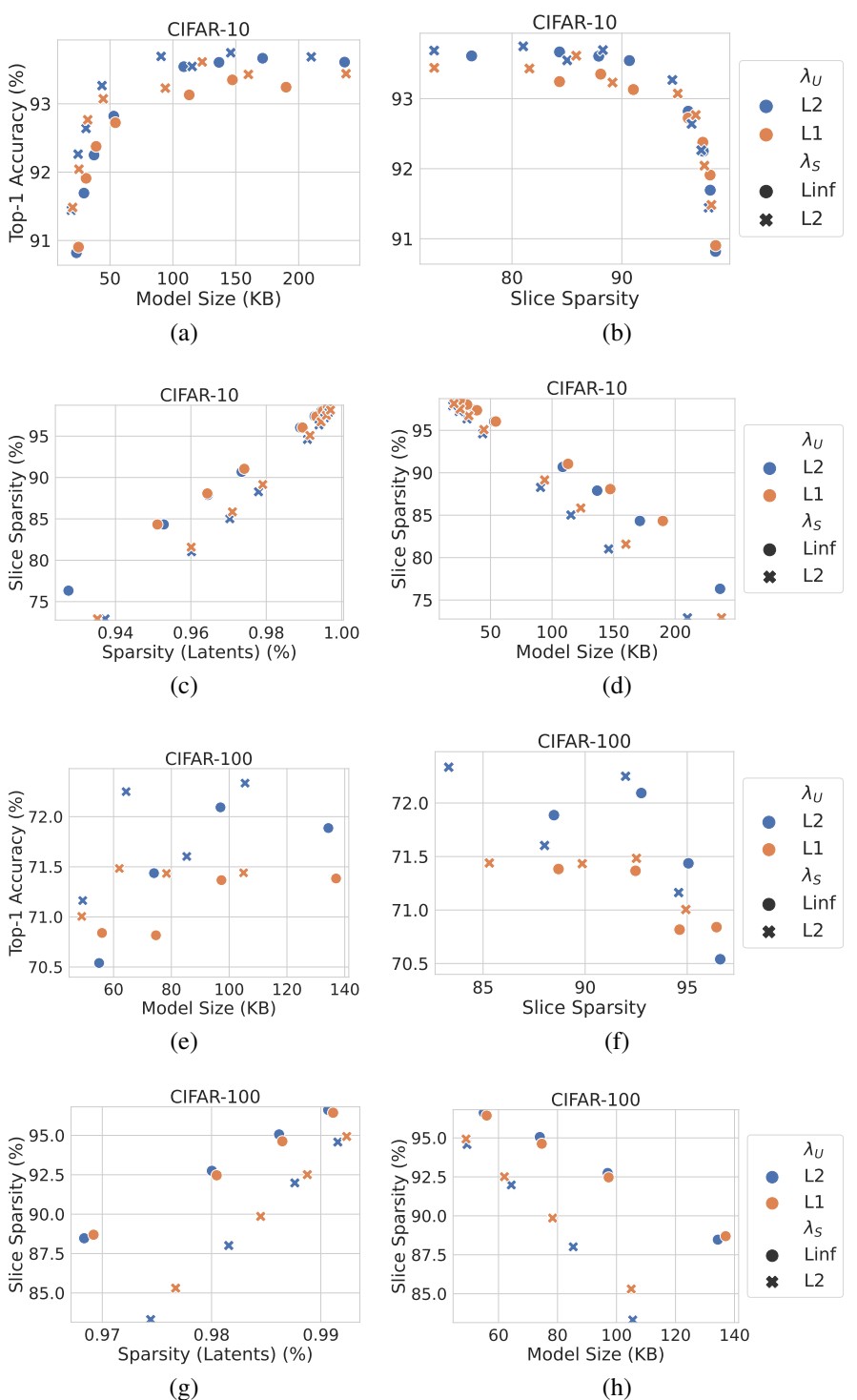

Figure 11: Comparison of $l_2$ *vs.* $l_1$*vs.* $l_\infty$ norm for various metrics of sparsity and size for both CIFAR-10 (top row) and CIFAR-100 (bottom row). We see that $l_2$ group norm does better than $l_\infty$ group norm in terms of accuracy vs model-size or slice sparsity (a,b,e,f). $l_1$ weight norm has little additional effect compared to $l_2$ weight norm. $l_\infty$ favors higher slice sparsity for the same level of sparsity (c,g). $l_\infty$ tends to result in higher model size for a given slice sparsity but also higher slice sparsity given a model size, which shows the tradeoff between compression and sparsification.

