# OpenReview forum: "LilNetX: Lightweight Networks with EXtreme Model Compression and Structured Sparsification"
_ICLR.cc/2023/Conference — ICLR 2023 poster_

### Official Review · Reviewer_1dzT · 2022-10-21

**Confidence:** 4
**Clarity, Quality, Novelty And Reproducibility:** The paper is of sound clarity.
**Correctness:** 3
**Technical Novelty And Significance:** 2
**Empirical Novelty And Significance:** 2
**Recommendation:** 6

**Strength And Weaknesses:**

Strength:
The authors show that their approach can give a significantly better storage size compression ratio when compared with earlier works.

Weakness:
Some statements in the manuscript are not well supported. Moreover, there is a lack of comparison with a proper baseline, especially for the acceleration experiments.

It is not true: "Most quantization works only optimize for model compression and not for inference speedups.". It is a standard industry practice to use quantized models, and the use case of an optimized model only for storage size compression does exist but is limited.

It is unclear how much of the performance gain is from the training procedure and fine-tuning. Additional ablation experiments would make the paper stronger.

The slice-sparsity approach can be considered a particular case of block sparsity, and I believe there is probably a need for additional justification for this new approach.

The uncompressed MobileNet-v2 accuracy presented in the paper is incorrect. It should be 28 instead of 32.7

**Summary Of The Paper:**

This work adopted a combination of losses that jointly regulate model size, weight decay, and structured sparsity to achieve reduced weight encoding bits and slice sparsity. The authors show promising results in reducing model storage size in their experiments.

**Summary Of The Review:**

Overall this work presented promising results but can use extra improvement.

---

> ### Author Response · Authors · 2022-11-14
> **Authors' response**
>
> Thank you for reviewing the paper and providing your feedback. We address the major concerns below:
>
> **1) Comparison for acceleration:**
>
> Most baselines utilizing unstructured sparsity [1] (LTH) or integer quantization [2] require dedicated hardware for obtaining inference speedups and have specific setups making it difficult to provide apple-to-apple comparison on a standard benchmark. We show results for acceleration to highlight the efficacy of our approach to obtain inference speedups even with additionally optimizing for model size, which is not present in previous model compression algorithms [3].
>
> **2) Importance of model compression:**
> We acknowledge that quantization works can obtain inference speedups as is standard in industry but require specialized hardware as mentioned above. We have updated the text to reflect that.
> Model compression is still of importance in certain settings which can have low network bandwidth and memory capacity as also highlighted by prior work [3]. Large networks which are trained repeatedly and then transmitted to a number of storage-limited devices will thus require high levels of model compression. We agree that runtime of these networks are also important and hence optimize for inference speedups through structured sparsity.
>
> **3) Training:**
>
> We would like to highlight that our approach requires no fine-tuning and is fully end-to-end with a single training stage. The training follows the standard procedure of dense models along with the regularization constraints. We vary only the sparsity priors whose ablations are provided in Section 4.
>
> **4) Block sparsity:**
>
> Slice sparsity is indeed a special case of block sparsity. However, block sparsities of even 4x4 size consist of 16 elements or dimensions in the latent space. This however increases the model size due to storage of more dimensions compared to slice sparsity of a 3x3 kernel which consists of 9 dimensions. Additionally, block sparsity is achieved in the model space only by constraining all 16 elements in the latent space to be 0. This is a much stronger constraint than slice sparsity which results in performance drops even with low sparsity levels. Moreover, most GPUs require blocks of atleast 8x8 or 16x16 [4] to achieve speedups which quadratically scales the number of elements. We will include this in the final draft the paper. We are working on providing experiments for block sparsity as well for comparison with slice sparsity.
>
> **5) MobileNet accuracy:**
>
> The numbers for our MobileNet experiments correspond to training using the FFCV library for 35 epochs which obtained an error rate of 32.7%. We will include the results for longer training of MobileNet experiments in the paper as permitted by time and computation constraints. We train the sparse compressed and dense uncompressed models for an equal number of epochs to show that the sparse compressed models can perform equally well as dense models for the same training schedule and still outperform other baseline approaches.
>
> [1] Frankle, Jonathan, and Michael Carbin. "The lottery ticket hypothesis: Finding sparse, trainable neural networks." arXiv preprint arXiv:1803.03635 (2018).
>
> [2] Young, Sean I., et al. "Transform quantization for CNN compression." IEEE Transactions on Pattern Analysis and Machine Intelligence 44.9 (2021): 5700-5714.
>
> [3] Oktay, Deniz, et al. "Scalable model compression by entropy penalized reparameterization." arXiv preprint arXiv:1906.06624 (2019).
>
> [4] Gray, Scott, Alec Radford, and Diederik P. Kingma. "Gpu kernels for block-sparse weights." arXiv preprint arXiv:1711.09224 3 (2017): 2.

---

> > ### Author Response · Authors · 2022-12-06
> > **Comparison with block sparsity**
> >
> > We show comparison of our proposed slice sparsity with block sparsity when applied to the ResNet18 network trained on ImageNet. The blocks are 8x8/16x16 size which are the typical size supported by existing libraries [1]. As a fully learnable weight decoder introduces a large overhead for block sparsity (~256 kB for the case of 16x16 blocks), we use a fixed DFT matrix instead with learnable diagonal scaling coefficients (overhead of ~1 kB) as in [2]. Even with such a setup, we see that slice sparsity outperforms block sparsity with lower model size and higher sparsity. A larger drop in accuracy is observed with higher block size. This is in line with our hypothesis (mentioned above), where block sparsity results in higher model size due to higher dimensions of the latents. Additionally, sparsity is much lower as it requires constraining all 64 elements (for 8x8) or 256 elements (for 16x16) to be zero in the latent space in order to obtain sparsity in the model space.
> >
> >
> > | Sparsity type          | Size (MB)      | Error (Top-1 %) | Sparsity (%) |
> > |------------------------|----------------|-----------------|--------------|
> > | Slice Sparsity         | **1.58 (30x)** | **30.1**        | **33.3**     |
> > | Block Sparsity (8x8)   | 3.34 (14x)     | 30.1            | 8.4          |
> > | Block Sparsity (16x16) | 3.67 (13x)     | 30.8            | 12.8         |
> >
> > [1] Gray, Scott, Alec Radford, and Diederik P. Kingma. "Gpu kernels for block-sparse weights." arXiv preprint arXiv:1711.09224 3 (2017): 2.
> >
> > [2] Oktay, Deniz, et al. "Scalable model compression by entropy penalized reparameterization." arXiv preprint arXiv:1906.06624 (2019).

---

### Official Review · Reviewer_HHQc · 2022-10-25

**Confidence:** 4
**Correctness:** 3
**Technical Novelty And Significance:** 2
**Empirical Novelty And Significance:** 2
**Recommendation:** 6

**Clarity, Quality, Novelty And Reproducibility:**

Overall the paper is well written and easy to follow. The algorithm itself is of limited novelty (combining a prior based quantization approach from Oktay, 2020 ,and a common structured sparsity based loss.). However, I did not find any previous work which tackles model compression and structured sparsity simultaneously explicitly. The paper may therefore be of independent interest to some in the community. The authors have provided the setup and code to reproduce their results, which I was able to partially validate.

**Strength And Weaknesses:**

**Strengths**

1. The problem of model compression which preserving inference speed is important for realtime and edge applications, and therefore interesting to the community.
2. The authors show comparable results to existing compression approaches (Oktay et al. especially) while also adding sparsity as a constraint.
3. The algorithm cleverly uses a zero mean prior to further enforce sparsity.

**Weaknesses**
1. I am not sure why the loss contains both a group sparse and sparsity enforcing constraint. These are obviously analogous constraints, and ideally the algo should be able to achieve the same result by adjusting the structured sparsity constraint. Fig. 3 shows some effects of sparsity on the model size, but it does not seem to be due to any change in slice sparsity. Could the authors clarify what Fig.3 is supposed to represent?

2.  The paper also does not compare with more modern sparsification methods (for eg. Continuous Sparsification [1], structured continuous sparsification[2]) which achieve similarly sparse models while retaining performance. It would be fair to compare model sizes and accuracies of standard sparsification algorithms as well.

3. I also suggest the authors add bit-quantization based approaches to the comparisons so that their results can be placed in the context of common compression methods. A few simple baselines of 8-bit or 4-bit quantization could help readers validate the need for training models with the proposed algorithm.

Refs

[1] Savarese et al., Winning the Lottery with Continuous Sparsification

[2] Yuan et al., Growing Efficient Deep Networks by Structured Continuous Sparsification

**Summary Of The Paper:**

This paper proposes an algorithm to optimally compress a model while preserving performance and improving inference speed. The algorithm achieves compression by modelling the weights as quantized latent representations sampled from a Gaussian prior which are further optimized using an entropy penalty. In order to ensure inference speedup, the algorithm also leverages a structured sparsity based loss. The authors show the efficacy of their algorithm on CIFAR-10/100 and Imagenet and claim better model compression due to the analogous behavior of the two constraints.

**Summary Of The Review:**

Given the limited novelty and missing comparisons with modern structured sparsity enforcing algorithms, I am currently inclined towards a weak reject. The performance boosts due to the weight sampling mechanisms are unclear given that recent results in bit level compression have shown almost no loss in accuracy with significant speedups. However, i am open to changing my mind if the authors can provide reasonable rebuttals to my concerns.

---

> ### Author Response · Authors · 2022-11-14
> **Clarification of the two sparsity enforcing constraints and comparison with quantization and modern sparsification methods**
>
> Thank you for reviewing the paper and providing your feedback. We address the major concerns below:
>
> **1) Why does the loss contain two sparsity terms?**
>
> The two sparsity terms achieve different objectives and the end result (model size and slice sparsity) is not achievable by just the group sparsity term. Group sparsity encourages terms in a group to go to zero simultaneously (but doesn't encourage sparsity within a group), while unstructured sparsity terms introduces sparsity within a group as well. This is shown in Eq. 3 of [1] where the two terms achieve the mixed effect of unstructured sparsity (within group sparsity) and structured/slice sparsity (group sparsity). Unstructured sparsity results in lower model size as more zeros in the quantized latent space lead to low entropy.
> This is evident in Fig. 3(a), where a high $\lambda_U$ gives lower model size at the same accuracy (green line vs orange/blue line). Fig. 3(b) shows a marginal gain in slice sparsity with higher $\lambda_U$. This is expected as the unstructured sparsity prior does not directly optimize for slice structured sparsity. Fig. 3’s key takeaway is that $\lambda_U$ shows a strong improvement in model size while also marginally benefitting sparsity. Similarly, Fig. 4 shows that $\lambda_S$ improves slice sparsity while marginally improving model size. Both coefficients are necessary to optimize for the dual metrics of model size and sparsity.
>
> **2) Additional Sparsification Baselines**
>
> We thank the reviewer for bringing these works to our notice. While we had included comparison with Frankle et al’s LTH work, [2] and [3] are indeed relevant works on sparsification of neural networks, and we will add their reported numbers to the Table 1 of the main paper. Note that, generally speaking, while sparsity methods obtain very high accuracy values, they often do not achieve high compression (because of floating point weights) and end up with relatively larger model size. We include comparisons with these approaches in the table below. We include model size for sparsity works based on storing only non zero weights and not the overhead of storing their index locations as well.
>
> **3) Additional Quantization Baselines**
>
> We had included [4] in Table 1, which performs quantization and achieves low bitrates but still does not achieve high compression levels compared to ours. While quantization based methods achieve speedups, they require specialized hardware for low bit-depth arithmetic [4,5,7]. Our method proposes to use floating points (sparse) weights during inference and can run on commodity hardware. We include comparisons with other quantization approaches [6,7] and summarize the results in the table below. [6] require post-hoc training stages to improve performance after quantization, unlike our approach (single training stage). We obtain high levels of compression rates compared to [6,7] at similar levels of accuracy rates.
>
> Table: Baseline comparison for ResNet-50 trained on ImageNet. *refers to unstructured sparsity
> | Method                              | Size (MB)   | Error (Top-1 %) | Sparsity |
> |-------------------------------------|-------------|-----------------|----------|
> | Uncompressed                        | 102.00 (1x) | 23.7            | 0.0      |
> | Savarese et al. [2]                 | 8.36 (12x)  | 24.5            | 91.8*    |
> | Yuan et al. [3]                     | 38.76 (3x)  | 24.8            | 38.8     |
> | Jain et al. [6] (4 bit)             | 12.75 (8x)  | 25.6            | 0.0      |
> | Jain et al. [6] (8 bit, no retrain) | 25.5 (4x)   | 25.7            | 0.0      |
> | Jain et al. [6] (8 bit)             | 25.5 (4x)   | 24.6            | 0.0      |
> | Zhao et al. [7] (4 bit)             | 12.75 (8x)  | 33.8            | 0.0      |
> | Zhao et al. [7] (8 bit)             | 25.5 (4x)   | 25.3            | 0.0      |
> | LilNetX (Best)                      | 3.94 (26x)  | 25.4            | 66.7     |

---

> > ### Author Response · Authors · 2022-11-14
> > **References**
> >
> >
> > [1] Simon, Noah, et al. "A sparse-group lasso." Journal of computational and graphical statistics 22.2 (2013): 231-245.
> >
> > [2] Savarese, Pedro, Hugo Silva, and Michael Maire. "Winning the lottery with continuous sparsification." Advances in Neural Information Processing Systems 33 (2020): 11380-11390.
> >
> > [3] Yuan, Xin, Pedro Savarese, and Michael Maire. "Growing efficient deep networks by structured continuous sparsification." arXiv preprint arXiv:2007.15353 (2020).
> >
> > [4] Young, Sean I., et al. "Transform quantization for CNN compression." IEEE Transactions on Pattern Analysis and Machine Intelligence 44.9 (2021): 5700-5714.
> >
> > [5] Han, Song, et al. "EIE: Efficient inference engine on compressed deep neural network." ACM SIGARCH Computer Architecture News 44.3 (2016): 243-254.
> >
> > [6] Jain, Sambhav, et al. "Trained quantization thresholds for accurate and efficient fixed-point inference of deep neural networks." Proceedings of Machine Learning and Systems 2 (2020): 112-128.
> >
> > [7] Zhao, Ritchie, et al. "Improving neural network quantization without retraining using outlier channel splitting." International conference on machine learning. PMLR, 2019.

---

> > ### Comment · Reviewer_HHQc · 2022-11-30
> > **Thank you for the response**
> >
> > i thank the authors for answering my queries, and addressing the weaknesses. While I am still concerned about the novelty, the overall algorithm does appear to add value over structured sparsity based approaches in terms of compression. This may be of interest to many in the model compression community. I am, therefore, raising my score to 6.

---

> > > ### Author Response · Authors · 2022-12-05
> > > **Author response**
> > >
> > > Thank you for raising the score. Your feedback has been helpful in improving the paper and we will include the changes in the final draft.

---

### Official Review · Reviewer_TxzT · 2022-10-28

**Confidence:** 4
**Clarity, Quality, Novelty And Reproducibility:** Na
**Correctness:** 3
**Technical Novelty And Significance:** 3
**Empirical Novelty And Significance:** 2
**Recommendation:** 6

**Strength And Weaknesses:**

Strength:

The paper is well-written, and code is also provided for reproductivity.

Compared to the previous pruning method, this paper is more practical, because mode deployment needs to consider model size and memory.

This paper analysis part is Thorough.




Weaknesses:

In terms of the model's size, the author are encouraged to discuss different sparse formats (CSR, COO, N:M) [1] to encode the non-zero elements.





[1]. STen: An Interface for Efficient Sparsity in PyTorch


**Summary Of The Paper:**

This paper proposes a new framework to jointly optimize the model size and inference acceleration.  Compared to previous methods, This paper can train the sparse model considering the model size and computation efficiency simultaneously using an end-to-end manner. The unified framework is novel and elegant.

**Summary Of The Review:**

Na

---

> ### Author Response · Authors · 2022-11-14
> **Effect of compression using sparse matrix formats**
>
> Thank you for reviewing the paper and providing your feedback. We are pleased to see that you found our paper well-written, practical and thorough.
>
> We thank you for pointing out the different sparse formats which can be used to encode the non-zero elements of the quantized latent representations. We have included an analysis subsection in section C of the appendix which compares the performance of the various sparse matrix formats along with arithmetic coding in terms of model size. We have also included the table below for ease of readability.
>
> We choose the two popular formats of Compressed Sparse Row (CSR) or Coordinate Format (COO). Results are summarized in the table below for our best run for ResNet-50 shown in Table 1 in the main paper. We see that entropy coding far outperforms the sparse formats of CSR and COO with COO obtaining better compression rates than CSR. This is expected as CSR/COO achieves high levels of compression only with extremely high levels of sparsity. With an unstructured sparsity level of ${\sim}$80\%, storing just the non zero weights itself (and not their indices) provides a maximum compression of $5\times$.
>
> Model size with different types of compression methods of the quantized latents:
> | Entropy Coding | CSR     | COO     | Slice Sparsity (%) | Unstructured Sparsity (%) |
> |----------------|---------|---------|--------------------|---------------------------|
> | **3.96 (26x)** | 57 (2x) | 30 (3x) |               66.7 |                      78.8 |

---

### Official Review · Reviewer_gqW8 · 2022-11-02

**Confidence:** 4
**Clarity, Quality, Novelty And Reproducibility:** 1) Clarity. As I mentioned above, the…
**Correctness:** 4
**Technical Novelty And Significance:** 3
**Empirical Novelty And Significance:** 4
**Recommendation:** 8

**Strength And Weaknesses:**

Pros.
1) I think the core idea is interesting and reasonable: building a latent space of quantized model parameters for simultaneously learning model compression and sparsity (both unstructured and structured) with continuous weights maintained for sparsity learning.
2) Experimental results on CIFAR-10 and ImageNet in Table 1 are impressive. The proposed method achieves much better compared to previous methods, from perspective of both compression rate and accuracy.

Cons.
My major concern is that the whole paper (including both the method description and experiments) does not clearly introduce contribution of quantization to the proposed method. To be specific:
1) After reading Section 3, I am still not clear how the model weights are quantized during training. The Equation (8) does not show processing on quantization.
2) In Section 4 and Section 5, the role/contribution of quantization to the experimental results is not formally analyzed. Maybe ablation studies are needed.

The paper clearly describes unstructured and structured sparsity but not quantization. According to my experiences on DNN model compression, quantization performs well on both model compression and acceleration, while structured spasity performs better on acceleration compared to unstructured sparsity. Anyway, I am not clear about the specific contribution of the latent space of quantized weights.

**Summary Of The Paper:**

The paper proposes to simultaneously learn model compression and sparsity in a latent space. Experiments on CIFAR-10 and ImageNet show that the proposed methods significantly outperforms previous competitive methods on compression and pruning only.

**Summary Of The Review:**

The code idea is novel and experimental results are impressive, though contributions of the latent space of quantized weights are not clearly introduced.

---

> ### Author Response · Authors · 2022-11-14
> **Clarification of quantization and its role**
>
> Thank you for reviewing the paper and providing your feedback. We are delighted to see that you found our ideas interesting and reasonable, and our experimental results impressive. We address the major concerns below:
>
> **1) How are the model weights quantized?**
>
> For every weight $\boldsymbol{W}$ in the model space, we have corresponding latent space quantized weights $\widetilde{\boldsymbol{W}}$. However, to make the quantized weights learnable we maintain continuous surrogates $\widehat{\boldsymbol{W}}$. *The quantized weights are obtained by rounding the continuous surrogates to the nearest integer*. Thus we perform scalar quantization for each element of the continuous surrogate. We use a straight-through estimator to make the rounding operation differentiable. The quantized weights are then passed to the decoder to obtain the weights in the model space $\boldsymbol{W}$ which are continuous in nature. After training we keep only the quantized weights and discard the continuous surrogate values. Note that the model’s convolutions use the continuous weights in the model space $\boldsymbol{W}$ and inference speedups are obtained due to the presence of sparsity in these weights. The benefit of the quantized weights is that they can be efficiently stored using arithmetic coding and lead to large reductions in model size. At inference time, the quantized weights are decoded using the decoder to obtain the continuous weights.  We provided a brief explanation of this process in the paragraph before and after Eq. 3 of Section 3.1. We have added further explanation to the text to make it clearer.
>
> **2) Role of quantization**
>
> Quantization is necessary for entropy coding, which converts a sequence of quantized values to a stream of bits. Quantized weights with lower entropy lead to lower model size. Section 4 discusses the accuracy vs. model size trade-off (albeit via varying the sparsity coefficients). We have also included histograms of the quantized weights for different layers in the supplementary (section B) to better visualize the quantized latent space. As an additional ablation, we include the effect of using a fixed DFT transform matrix with learnable diagonal scaling coefficients as our decoder instead of a decoder matrix with all learnable elements. The results are shown in the table below. We see that both types of decoder matrices obtain similar results with DFT obtaining slightly better numbers in terms of model size and sparsity for ResNet-50.
>
> | Network   | Decoder   | Size (MB)      | Error (Top-1 %) | Sparsity (%) |
> |-----------|-----------|----------------|-----------------|--------------|
> | ResNet-18 | Learnable | **1.58 (30x)** | 30.1            | 33.3         |
> | ResNet-18 | DFT       | 1.84 (25x)     | 30.1            | **36.2**     |
> | ResNet-50 | Learnable | 3.94 (26x)     | 25.4            | 66.7         |
> | ResNet-50 | DFT | **3.81 (27x)** | 25.4            | **70.5**     |
>
> We thank you again for your valuable suggestions and positive score. This will indeed improve our submission and we will add it to the main manuscript in the final version for the benefit of the readers.

---

> > ### Comment · Reviewer_gqW8 · 2022-12-10
> > **I am satisfied with the authors' answers**
> >
> > I am satisfied with the authors' answers. I will keep my score as "8: accept, good paper".

---

> > > ### Author Response · Authors · 2022-12-10
> > > **Author response**
> > >
> > > Thank you for the positive score and your feedback. We will include the changes in the final draft.

---

### Author Response · Authors · 2022-12-06
**Summary of proposed changes**

Dear reviewers and AC,

We thank you for your time and valuable feedback. We also thank reviewer HHQc for increasing their score. We believe the reviews have definitely helped us improve our submission.

We will incorporate the following changes in the manuscript as discussed with the reviewers. Some changes are highlighted in blue in the current draft.

1. Include a more detailed explanation of quantization and its effects in Section 3.1.
2. Visualize the weight histograms of the quantized latents.
3. Add a comparison of different sparse matrix formats with entropy coding in terms of the compression rate.
4. Include additional comparisons with pruning and quantization approaches in Table 1 and update related works.
5. Include comparison and an explanation of why slice sparsity is more beneficial for our approach than standard block sparsity.

---

### Decision · Program_Chairs · 2023-01-20

**Decision:**

Accept: poster

**Justification For Why Not Higher Score:**

Three reviewers agree that the paper is marginally above the accept line, while one reviewer likes the latent space modeling quantization idea and thinks it a good paper. So it is a solid poster paper.

**Justification For Why Not Lower Score:**

The reviewers unanimously suggest to accept the current manuscript, so is the decision.

**Metareview: Summary, Strengths And Weaknesses:**

The manuscript proposes to simultaneously learn model compression and sparsity in a latent space. The algorithm achieves compression by modelling the weights as quantized latent representations sampled from a Gaussian prior which are further optimized using an entropy penalty. Such a latent space of quantized model parameters is used for simultaneously learning model compression and sparsity (both unstructured and structured) with continuous weights maintained for sparsity learning. The authors show the efficacy of their algorithm on CIFAR-10/100 and Imagenet. Reviewers unanimously agree that the current manuscript is above the accept line, so is the decision.

**Note From Pc:**

if the above contains the word "oral" or "spotlight" please see: "oral" presentation means -> notable-top-5% and "spotlight" means -> notable-top-25%. As stated in our emails, we are disassociating presentation type from AC recommendations